# Fasting and Postprandial DNA Methylation Signatures in Adipose Tissue from Asymptomatic Individuals with Metabolic Alterations

**DOI:** 10.3390/ijms262311306

**Published:** 2025-11-22

**Authors:** Fabiola Escalante-Araiza, Angélica Martínez-Hernández, Humberto García-Ortiz, Eira Huerta-Ávila, José Rafael Villafan-Bernal, Cecilia Contreras-Cubas, Federico Centeno-Cruz, Edna J. Nava-González, José Damián Carrillo-Ruiz, Ernesto Rodriguez-Ayala, Raúl A. Bastarrachea, Francisco Barajas-Olmos, Lorena Orozco

**Affiliations:** 1Immunogenomics and Metabolic Diseases Laboratory, Instituto Nacional de Medicina Genómica, Secretaría de Salud, Mexico City 14610, Mexico; fabiola.escalante@anahuac.mx (F.E.-A.); amartinez@inmegen.gob.mx (A.M.-H.); hgarcia@inmegen.gob.mx (H.G.-O.); lneeha920609@gmail.com (E.H.-Á.); joravibe@hotmail.com (J.R.V.-B.); ccontreras@inmegen.gob.mx (C.C.-C.); fcenteno@inmegen.gob.mx (F.C.-C.); 2Faculty of Health Sciences, Universidad Anáhuac Norte, Mexico City 52786, Mexico; ernesto.rodriguez@anahuac.mx; 3Investigador por México, Secretaría de Ciencia, Humanidades, Tecnología e Innovación (SECIHTI), Mexico City 14610, Mexico; 4Facultad de Salud Pública y Nutrición, Universidad Autónoma de Nuevo León, Monterrey 64460, Mexico; edna.navang@uanl.edu.mx; 5Faculty of Psychology, Universidad Anáhuac Norte, Mexico City 52786, Mexico; damian.carrillo@anahuac.mx; 6Stereotactic and Functional Neurosurgery, Hospital General de Mexico, Mexico City 06720, Mexico; 7Population Health Program, Texas Biomedical Research Institute, San Antonio, TX 78227, USA; raul@txbiomed.org

**Keywords:** subcutaneous adipose tissue, postprandial state, mixed meal challenge, obesity, prediabetes, type 2 diabetes, epigenetic biomarkers, differential DNA methylation

## Abstract

Cardiometabolic phenotypes such as obesity and impaired insulin action are key determinants of type 2 diabetes (T2D). Growing evidence highlights the postprandial state as a critical window in metabolic regulation, where epigenetic mechanisms, particularly DNA methylation in insulin-sensitive tissues, may play pivotal roles. However, their dynamics across prandial states in subcutaneous adipose tissue (SAT) remain unclear. We analyzed genome-wide DNA methylation in paired fasting and postprandial SAT biopsies from 29 asymptomatic, drug-naïve individuals classified as controls (*n* = 8), prediabetes *n* = 9), or T2D (*n* = 12). Postprandial samples followed a standardized mixed-meal test. DNA methylation was quantified using the Illumina MethylationEPIC array and analyzed through the Chip Analysis Methylation Pipeline (ChAMP) pipeline. Differential methylation was more pronounced postprandially, especially in the T2D group. After adjusting for age and sex, 4599 differentially methylated CpG sites (DMCs) were identified, with increased hypermethylation in T2D. A total of 130 DMCs across 99 genes, including *LCLAT1*, *HLA-C*, *ZNF714*, and *HOOK2*, were shared by prediabetes and T2D groups. Over-representation analysis revealed 202 enriched pathways related to insulin resistance, AMPK signaling, and immune responses. Additionally, 110 Differentially Methylated Regions (DMRs), including *ZNF577* and *AGPAT1*, were detected. These findings reveal early, prandial-dependent epigenetic alterations in SAT that precede overt dysglycemia, offering insights into personalized prevention in T2D.

## 1. Introduction

Immunometabolic impairments such as insulin resistance, compensatory hyperinsulinemia, dysglycemia, dyslipidemia, and excess adiposity are well-established predictors of type 2 diabetes. The global burden of metabolic diseases is substantial; 77% of the 41 million deaths attributable to these non-communicable diseases annually occur in low- and middle-income countries [1]. Although several studies have investigated the biological mechanisms underlying metabolic and nutrition-related diseases, early inter-individual differences regarding susceptibility to metabolic dysregulation in specific tissues remain an emerging field of study. Most metabolic studies have primarily focused on the fasting state, despite growing evidence that certain postprandial biomarkers can predict cardiovascular risk [2,3,4,5].

Influenced by environmental factors, metabolic diseases are widely recognized to interact across various biological levels [6]. The epigenome plays a critical role in regulating cellular processes, such as gene expression, which is heritable independent of changes in the DNA sequence [7]. Epigenetic regulation occurs through various mechanisms, including DNA methylation, histone modification, chromatin remodeling, and noncoding RNA modulation. Among these epigenetic mechanisms, DNA methylation is one of the most extensively studied epigenetic markers [8,9,10,11,12]. Altered DNA methylation and the corresponding changes in gene expression have been related to obesity and glucose impairment in cross-sectional and longitudinal studies [13,14,15,16,17,18]. Our group previously identified several DNA methylation alterations in both adipose tissue and its mesenchymal stem cells during T2D, and these changes reverted after a clinical intervention [13,14,15]. Recent evidence suggests that methylation profiles in blood can differentiate between the fasting and postprandial states [19,20]. However, tissues targeted by insulin should be studied in depth by explaining the differences among endophenotypes of patients with diabetes.

To improve prevention strategies for cardiovascular risk phenotypes of immunometabolic origin, it is imperative to investigate short- and long-term epigenetic modifications as key factors influencing metabolic processes in both the fasting and postprandial states in insulin-targeted tissues, such as adipose, liver, and muscle. The present study investigated DNA methylation in subcutaneous adipose tissue (SAT) samples from asymptomatic drug-naïve individuals with impaired glucose metabolism compared with individuals with normal blood glucose levels, in both the fasting and postprandial states.

## 2. Results

### 2.1. Demographic Characteristics of the Sample

The present study aimed to investigate altered DNA methylation in SAT from drug-naïve, symptom-free, metabolically impaired individuals with prediabetes (PD) and T2D compared to a control group, as well as to analyze differences in DNA methylation between two prandial states in each group: fasting and postprandial. This study included samples and SAT samples from the GEMM family study (Genética de las Enfermedades Metabólicas en México/Genetics of Metabolic Diseases in Mexico Family Study). Significant differences between the control group and PD and T2D groups were observed for all clinical traits. The greatest alterations in all cardiometabolic profiles were observed in the T2D group (Table 1).

### 2.2. Differential DNA Methylation

Global DNA methylation data from SAT taken during the fasting and postprandial states were normalized and subjected to quality control. We identified 746,969 CpG sites. Hierarchical clustering analysis revealed distinct grouping patterns for the control, prediabetes, and T2D groups, displaying variations between both prandial states (Figure 1). In the fasting state, controls were distributed across three clades; in the postprandial state, they clustered together in a single clade. In contrast, the prediabetes and T2D groups were mixed in both prandial states, though most cases were found in the same clade in the postprandial state.

#### 2.2.1. Differential DNA Methylation by CpG Sites

To identify DMCs associated with metabolic impairment, we compared β-values between cases (prediabetes or T2D) and controls by prandial state. A schematic overview of the group contrasts is provided in Appendix A. We identified 4599 DMCs with an adjusted *p*-value < 0.05 and |∆β| > 10% (Appendix A). Notably, the T2D group had nearly twice the number of DMCs in the postprandial state than the fasting state (1829 vs. 832), whereas the prediabetes group had a similar number of DMCs in both prandial states (970 vs. 968; Appendix A). As expected, both the prediabetes and T2D groups exhibited predominantly hypermethylated DMCs in the postprandial state, with the T2D group having the highest percentage (88.24%) of hypermethylation (Figure 2) and the highest number of exclusive genes and DMCs (T2D: postprandial *n* = 1109, fasting *n* = 239; prediabetes: postprandial *n* = 383, fasting *n* = 303). Notably, we identified 130 shared DMCs in 99 genes that were consistent in the prediabetes and T2D groups in both the fasting and postprandial states (Appendix A). The top 15 DMCs, either hypermethylated or hypomethylated were found in lysocardiolipin acyltransferase 1 (*LCLAT1*), human leukocyte antigen-C (*HLA-C*), family with sequence similarity 53 member B (*FAM53B*), ninjurin 2 (*NINJ2*), *LOC100049716*, decaprenyl diphosphate synthase subunit 1 (*PDSS1*), exosome component 10 (*EXOSC10*), hook microtubule tethering protein 2 (*HOOK2*), dachsous cadherin-related 1 (*DCHS1*), carnosine dipeptidase 2 (*CNDP2*), FSHD region gene 1 family member B (*FRG1BP*), spermatogenesis and centriole associated 1-like *(C21orf56/SPATC1L*), and zinc finger protein 714 (*ZNF714*) (Appendix A).

To identify altered biological pathways linked to DMCs, we performed a KEGG pathway enrichment analysis in each contrast. We identified 102 significant unique terms (*p*-value < 0.05) across groups by prandial state (Appendix A). Among all comparisons, the T2D group in the postprandial state had the highest number of enriched (*n* = 70) and exclusive (*n* = 25) pathways.

In the fasting state, the most significant enriched pathways in the prediabetes group were type I diabetes mellitus (hsa04940), allograft rejection (hsa05330), and autoimmune thyroid disease (hsa05320), whereas the most significant enriched pathways in the T2D group were viral myocarditis (hsa05416), glycosaminoglycan biosynthesis (hsa00532), and human papillomavirus infection (hsa05165). In the postprandial state, the top pathways in the prediabetes group were type 1 diabetes mellitus (hsa04940), phagosome (hsa04145), and human papillomavirus infection (hsa05165), whereas the top pathways in the T2D group were AMPK (hsa04152), cGMP-PKG (hsa04022), and axon guidance (hsa04360). The top 25 enriched pathways for each contrast are presented in Figure 3. Notably, though the term type 2 diabetes (hsa04930) was not included in the 25 top pathways, it was only identified in the T2D contrast in both prandial states.

We identified 27 shared pathways in at least three contrasts. These pathways include key metabolic pathways, such as type 1 diabetes mellitus (hsa04940), Th1 and Th2 cell differentiation (hsa04658), insulin signaling pathway (hsa04910), PI3K-Akt signaling pathway (hsa04151), longevity regulating pathway (hsa04211), thermogenesis pathway (hsa04714), Notch signaling pathway (hsa04330), and thyroid hormone synthesis (hsa04918), as well as glycosaminoglycan biosynthesis (hsa00534) and other pathways related to the immune system and viral infections (Appendix A).

#### 2.2.2. Differentially Methylated Regions

Given that multiple DMCs were revealed in the same loci, DMRs were investigated across both groups and prandial states. We identified 155 DMRs at 33 genes: 73 in the T2D group in the postprandial state, 33 in the T2D group in the fasting state, 27 in the prediabetes group in the fasting state, and 22 in the prediabetes group in the postprandial state. Some were shared across groups and prandial states; five were revealed in all contrasts [chr19:52390810-52391789 (*ZNF577*), chr5:135415693-135416613 (*VTRNA2-1*), chr5:178986131-178986906 (*RUFY1/RABIP4*), chr6:32144978-32146779 (*AGPAT1* and *RNF5*) and chr6:152125861-152126938 (*ESR1*)]. The KEGG enrichment analysis did not show significant results with these genes (Appendix A).

### 2.3. Alteration of Differential DNA Methylation After Mixed-Meal Intake

We further investigated differences in DNA methylation between both prandial states within each group (control postprandial vs. control fasting, prediabetes postprandial vs. prediabetes fasting, and T2D postprandial vs. T2D fasting). Hierarchical clustering showed no distinct aggregation by prandial state, as samples tended to cluster individually. We compared average β-values between prandial states using a less stringent cut-off than previous comparisons (unadjusted *p*-value < 0.05 and |∆β| > 5%; Appendix A). The most differences between prandial states were observed for the control group, with 488 DMCs compared to 157 DMCs in the prediabetes group and 170 DMCs in the T2D group (Appendix A). Body, intergenic, and TSS1500 regions were the top three gene annotations among the groups. Interestingly, when comparing the methylation percentages among the three groups by prandial state, a progressive and increased shift in hypermethylation was observed, from 2.87% in the control group to 34.39% and 55.88% in the prediabetes and T2D groups, respectively (Figure 4).

Furthermore, we found 23 shared genes among groups with DMCs; SH3 and multiple ankyrin repeat domains 2 (*SHANK2*) was the only gene shared among all groups. KEGG enrichment analysis revealed 20 significantly enriched pathways in the control group, 5 in the prediabetes group, and 11 in the T2D group (Appendix A). The NOTCH signaling pathway (hsa04330) was the only pathway shared between the prediabetes and T2D groups, with mastermind-like transcriptional coactivator 2 (*MAML2*) as the only shared gene.

Finally, as several DMRs were identified among contrasts, we analyzed whether DMRs were present in the different prandial states. The T2D group was the only group with significant DMRs and annotated genes, including solute carrier family 25 member 24 (*SLC25A24*, chr1:108735312-108735719) and zinc finger protein 57 (*ZFP57*, chr6:29648161-29649092) (Appendix A).

## 3. Discussion

Cardiovascular risk phenotypes of immunometabolic origin are characterized by significant subclinical dysfunction influenced by both genetics and environmental factors, such as diet, exercise, circadian rhythms, and medication. Epigenetic regulation plays a relevant role in this process and can take place through various mechanisms, including DNA methylation, histone modifications, chromatin remodeling, and noncoding RNA modulation. Emerging evidence indicates that epigenetic regulation largely contributes to the occurrence and progression of multiple metabolic diseases [8]. Among the epigenetic mechanisms, DNA methylation has been suggested as a major driver in the environmental and epigenome intersection [21]. Furthermore, DNA methylation analysis using microarray technology in large cohort studies has generated substantial evidence linking methylation patterns to a wide range of diseases and phenotypic traits [22]. Together, these findings underscore the central role of DNA methylation as a robust and informative epigenetic marker for investigating metabolic disease risk. Although DNA methylation is generally viewed as stable, methylation profiles in blood have been suggested to vary in different prandial states among groups with different BMI [19]. Significant clinical interventions and lifestyle changes can lead to epigenetic remodeling, which is associated with metabolic alterations in insulin-targeted tissues [13]. However, most studies comparing DNA methylation profiles have primarily focused on DMCs in the fasting state, particularly in individuals with well-established glycemic alterations who are undergoing treatment for glycemic control [5,14,15]. Despite these findings, the inclusion of suitable control groups is essential in epigenome-wide association studies to distinguish between epigenetic changes associated with diseases and those arising from other confounding variables [23]. Furthermore, a critical gap remains in understanding the epigenetic changes that occur in insulin-targeted tissues during the transition from fasting to the postprandial state.

This study aimed to elucidate DNA methylation signatures in SAT from asymptomatic and drug-naïve individuals, comparing individuals with impaired glucose metabolism (prediabetes and T2D) to individuals with normal glucose levels in both the fasting and postprandial states. A key finding was the identification of distinct DNA methylation signatures between the control, prediabetes, and T2D groups, with substantial differences in both prandial states. Furthermore, hierarchical clustering analysis revealed that the control group formed a distinct cluster solely in the postprandial state, whereas the prediabetes and T2D groups exhibited overlapping patterns across both prandial states. This suggests that individuals with impaired glucose metabolism exhibit more variability in DNA methylation patterns regardless of whether they are in a fasting or postprandial state.

A total of 4599 DMCs in 1611 genes were identified in this study. The number of DMCs in the postprandial state was markedly higher in the T2D group (*n* = 1829) than in the prediabetes group (*n* = 970), suggesting that postprandial changes in DNA methylation may be more strongly linked to metabolic dysfunction in individuals with advanced stages of glucose impairment. Moreover, in the postprandial state, controls had greater hypomethylation, whereas the T2D group predominantly exhibited hypermethylation, with patients with prediabetes passing through an evident transition towards a hypermethylated state. These findings suggest that DNA methylation increases in glucose-impaired groups, particularly in the postprandial state and in specific CpGs.

Compared to the control group, we detected 130 DMCs that were shared between the prediabetes and T2D groups (Appendix A), with *LCLAT1*, *HLA-C*, *NINJ2*, *ZNF714*, *CNDP2*, and *HOOK2* among the top differentially methylated genes. *ZNF714*, which has emerged as a potential factor in metabolic diseases, as Crujeiras et al. reported differential DNA methylation in adipose tissue of individuals with insulin resistance, suggesting a potential role in the epigenetic regulation of metabolic pathways [24]. *LCLAT1* (Lysocardiolipin acyltransferase 1), also known as ALCAT1, is a key enzyme in phospholipid remodeling that directly impacts mitochondrial function and oxidative stress, processes central to metabolic disease, which has recently been suggested to play a role in diabetes and obesity through epigenetic mechanisms [25,26]. Carnosine dipeptidase II (CNDP2) has been described as catalyzing the condensation reaction of lactic acid and phenylalanine, a metabolite that has been associated with the amount of adipose tissue in humans [27]. Individuals with obesity and type 2 diabetes have reported abnormal intragenic DNA methylation of *HOOK2* when compared to individuals with non-diabetic conditions. The *HOOK* family is a group of cytoplasmic linker proteins associated with microtubules, possibly participating in GLUT4 translocation and glucose uptake [28].

In this same group of 130 DMCs, we also report five genes (*ABLIM1*, *S100P*, *ZNF516*, *RPTOR*, and *SLC43A2*) with previous reports of altered DNA methylation following caloric intake that also correlated with changes in expression levels [29]. S100P serum levels have been proposed as an indicator of peripheral neuropathy in type 2 diabetes [30]. ZNF516 has been implicated in the browning of adipose tissue [31]. In addition, RPTOR (Regulatory Associated Protein Of mTOR Complex 1) has been reported to play a key role in nutrient and insulin-sensing pathways regulating cell growth in cancer, while *SLC43A2* has also been identified as a regulator of the mTORC1 complex in cancer studies [32]. Taken together, these findings indicate the need to investigate this signaling axis in non-cancer metabolic contexts, particularly in adipose tissue and its relationship with metabolic diseases.

These genes could be potential targets of future investigations, which could focus on their role in adipose tissue function, insulin resistance, and inflammatory processes characteristic of T2D. We also identified 102 enriched KEGG pathways among the four contrasts (Appendix A). All contrasts had pathways related to insulin resistance and immune regulation, underscoring the complex interplay between metabolic and immune dysfunction in the development of T2D [33]. Notably, pathways related to type 1 diabetes (hsa04940), antigen processing (hsa04612), and autoimmune thyroid disease (hsa05320) were significantly enriched in the prediabetes group, whereas AMPK signaling (hsa04152), cGMP-PKG signaling (hsa04022), and axon guidance (hsa04360) were discovered in the T2D group. These findings are in line with inflammation, insulin resistance, and oxidative stress being the central drivers of disease progression, as is currently considered to be the case in metabolic diseases [34,35]. Interestingly, the pathway analysis also revealed viral infection-related and autoimmune disease pathways, suggesting a potential link with altered immune responses in the development of metabolic dysfunction through epigenetic remodeling.

The overlap of several pathways, such as insulin, NOTCH, and PI3K-Akt signaling, across the prediabetes and T2D groups could indicate that these pathways play a fundamental role in the metabolic dysregulation observed in both groups with impaired glucose metabolism. Dysregulation of these pathways has previously been linked to insulin resistance and obesity [36,37]. These findings support canonical T2D pathways being altered in DNA methylation signatures from asymptomatic individuals.

The identification of DMRs adds another layer of insight into the epigenetic changes occurring in response to metabolic alterations. DMRs were enriched in *ZNF577*, *VTRNA2-1*, *ESR*, *RUFY1/RABIP4, RNF5*, and *AGPAT1*, and some were shared across both prandial states. These regions may represent critical loci for regulatory changes that influence gene expression in metabolic tissues. For example, *AGPAT1* plays a role in lipid metabolism and its dysregulation could contribute to the altered adipose tissue function observed in metabolic diseases [38,39]. Moreover, *ZNF577* hypermethylation may serve as an epigenetic marker of obesity-related breast cancer and appears to be influenced by dietary patterns [40]. The presence of DMRs in *ESR1* and *RUFY1/RABIP4* suggests that these loci might be involved in the cellular responses to insulin and glucose, making them potential candidates for further research [41,42].

Our study provides valuable insights into the role of DNA methylation in adipose tissue during fasting and the postprandial state, but it is not without limitations. The sample size, even if adequate for an exploratory study, is relatively small, and the results may benefit from validation in larger cohorts, principally to identify alterations in DNA methylation after mixed-meal intake. In addition, the functional implications of the differentially methylated genes and pathways need to be explored through follow-up studies, including gene expression analyses and functional assays. Further research into the temporal dynamics of DNA methylation in response to diet, lifestyle, and pharmacological interventions will also be important for understanding the potential of epigenetic modifications as therapeutic targets.

## 4. Materials and Methods

### 4.1. Study Design

The GEMM family study [43] is a well-characterized multi-center cohort of symptom-free adults. Postprandial samples were obtained after a liquid mixed-meal challenge (macronutrient composition: 65% carbohydrate, 15% protein, and 20% fat). The biopsies were obtained after 3 h, a time point selected based on previous findings suggesting that adipose tissue is metabolically most active approximately 3 h post-caloric intake [44,45]. We previously characterized the postprandial adipose tissue dysfunction phenotype at a systemic and molecular level from a multiomics perspective. Thus, the detailed methodologies regarding meal challenge, tissue biopsies, and fasting and postprandial biochemical phenotyping were published previously [46]. This study was conducted in accordance with the principles of the Declaration of Helsinki (1975, revised in 2013), and the research protocol was reviewed and approved by the local Ethics Committee (approval date: 13 March 2018). All participants signed a consent form.

### 4.2. Participants

Anthropometric measurements were taken while the participants were fasting, as well as blood pressure and BMI were calculated. Waist circumference was measured in centimeters using a professional Gulick tape measure (North Coast Medical, Inc., Morgan Hill, CA, USA). Height was obtained using a portable HM200P Portstad stadiometer (Quick Medical, Issaquah, WA, USA) in meters. Weight was measured using a Tanita BC-418 body composition analyzer (Hanover, MD, USA) in kilograms. Body composition was assessed by dual-energy X-ray absorptiometry (Lunar Prodigy, GE Healthcare, Madison, WI, USA).

The individuals were classified into three groups according to glucose impairment. The control group (*n* = 8) was defined as BMI < 25 kg/m^2^, waist circumference < 90 cm for men or <80 cm for women, HbA1c < 5.7%, fasting glucose < 100 mg/dL, and 2 h postprandial glucose < 140 mg/dL. Glucose-impaired, newly diagnosed, drug-naïve participants were defined by the cut-off values for prediabetes and T2D in the current American Diabetes Association (ADA) guidelines [47]. Cases presented with one of three glucose alterations for prediabetes (*n* = 9) and T2D (*n* = 12): elevated fasting glucose, elevated postprandial glucose, or abnormal glycated hemoglobin. Obesity was defined as a BMI ≥ 30 kg/m^2^, and hypertension was defined as a systolic pressure ≥ 130 mmHg and a diastolic pressure ≥ 80 mmHg.

### 4.3. Fasting and Postprandial Phenotyping

After completing the anthropometric measurements, an intravenous line and catheter were placed in the median basilic vein of the forearm, and a fasting blood sample (time, 0 min) taken. Immediately after, the participant ingested the mixed meal as 30% of the total daily energy expenditure. Postprandial blood samples were taken every 15 min until the first postprandial hour was completed, and then at 90 min, 120 min, and 180 min. A SAT biopsy from the right thigh was then obtained by a surgeon, serving as the postprandial tissue biopsy 3 h after meal ingestion. Additional postprandial blood samples were taken at 240 min and 300 min. Immediately after, the intravenous catheter was removed and the participant given lunch and dismissed.

At the second visit, blood pressure and heart rate were recorded after a 12 h fast. A second SAT biopsy was then obtained from the left thigh. The participant was provided with breakfast and dismissed.

### 4.4. DNA Extraction and Quality Control

Both SAT biopsies were immediately placed in RNAlater^®^ solution (Ambion, Inc.; Invitrogen, Carlsbad, CA, USA) to preserve the integrity of the nucleic acids before being frozen and stored at −70 °C. DNA was extracted from 40–70 mg of SAT using the QIAamp DNA Mini Kit (Qiagen, Valencia, CA, USA) following the manufacturer’s protocol. DNA integrity was evaluated by 1% agarose gel electrophoresis and quantified by spectrophotometry using a NanoDrop ND-1000 Spectrophotometer (version 3.5.2, NanoDrop Technologies, Inc., Wilmington, DE, USA).

### 4.5. Methylation Microarray Data Analysis

A genome-wide DNA methylation analysis was performed with the Infinium Human MethylationEPIC BeadChip Kit (also known as the 850k) according to the manufacturer’s protocol (Illumina, San Diego, CA, USA). Raw data were extracted by GenomeStudio software (version 2011.1, Illumina). All samples passed standard quality controls. All computational and statistical analyses were performed in CRAN R project version 3.0. For DNA methylation analyses, iDAT files were processed in the ChAMP package. Raw β-methylation scores were determined by calculating the ratio of the intensity of the methylated probes to the total intensity, which was the sum of the methylated and unmethylated probe intensities. The β-methylation values ranged from 0 (unmethylated) to 100% (fully methylated). Filtering was performed according to ChAMP default parameters which involved removing probes with fewer than 3 beads in at least 5% of the samples, non-CpG probes, multi-hit probes, and Single-Nucleotide Polymorphism-related probes. In addition, sex chromosome probes were excluded, leaving 746,969 probes for further analysis. The data were then normalized using the beta-mixture quantile normalization method [48].

To identify potential sources of variation in the data set, we used the ChAMP singular value decomposition function, including sentrix ID, age, and gender as covariates. Probe annotations were based on the Human MethylationEPIC 1.0B5 manifest file.

To assess differences in methylation between groups, the mean β-value for each CpG site was calculated in each group of participants. The difference between groups was defined as |∆β|, which was calculated as the mean β-value of the prediabetes or T2D group minus the mean β-value of the control group. For the comparison between the prediabetes and T2D groups in the fasting and postprandial states, CpG sites with an adjusted *p*-value < 0.05 and |∆β| > 10% were considered DMCs. For the comparison between fasting and postprandial states, CpG sites with *p*-value < 0.05 and |∆β| > 5% were considered DMCs.

To identify DMRs (P_area_ < 0.05), we used the Bumphunter algorithm from the ChAMP library with the default parameters. This created clusters with a minimum of seven probes and a maximum separation gap of 300 bp to identify significant DMRs.

### 4.6. KEGG Pathway Analysis

We used the WEB-based Gene SeT Analysis Toolkit (WebGestalt) [49] to analyze gene overrepresentation. Gene symbols associated with DMCs and DMRs were assessed using the KEGG pathway identification analysis. All *p*-values were adjusted using the Benjamini–Hochberg method.

### 4.7. Statistical Analysis

Continuous values in clinical measurements are presented as median and interquartile range (IQR) unless otherwise indicated. Differences among groups were analyzed by Kruskal–Wallis and differences between cases and controls were analyzed by Dunn’s post hoc test for joint ranks. Statistical significance is represented as follows: * *p*-value < 0.05, ** *p*-value < 0.01, *** *p*-value < 0.001, and **** *p*-value < 0.0001. The calculations were performed using 2025 JMP^®^ 18 (Jmp Statistical Discovery LLC, Cary, NC, USA).

## 5. Conclusions

This study provides a comprehensive analysis of DNA methylation changes in SAT from individuals with metabolic alterations, highlighting the significance of both the fasting and postprandial states in shaping the epigenetic landscape. Moreover, the observation that individuals with T2D have more extensive epigenetic changes, particularly in the postprandial state, may reflect the heightened sensitivity of adipose tissue to dietary and metabolic fluctuations. Taken together, our data suggest that methylation precedes clinical manifestation independent of treatment; thus, our findings support the idea that epigenetic modifications in adipose tissue may serve as valuable biomarkers for early detection of disease progression and metabolic disturbances. This has the potential to be crucial in the management of prediabetes and T2D, presenting novel opportunities for timely intervention and the development of personalized treatment strategies. Further studies are needed to validate these findings and explore their functional relevance, to deepen our understanding of the mechanisms underlying the pathophysiology of prediabetes and its progression to T2D, and to improve strategies for the prevention and treatment of these metabolic diseases.

## Figures and Tables

**Figure 1 ijms-26-11306-f001:**
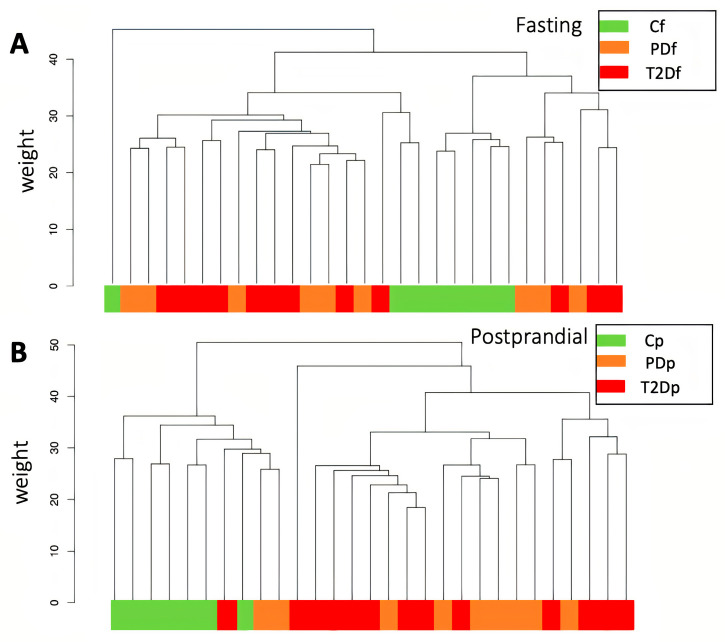
Hierarchical clustering dendrograms of DNA methylation in the fasting (**A**) and postprandial (**B**) states. Cf, control group-fasting; PDf, prediabetes group-fasting; T2Df, type 2 diabetes group-fasting; Cp, control group-postprandial; PDp, prediabetes group-postprandial; T2Dp, type 2 diabetes group-postprandial.

**Figure 2 ijms-26-11306-f002:**
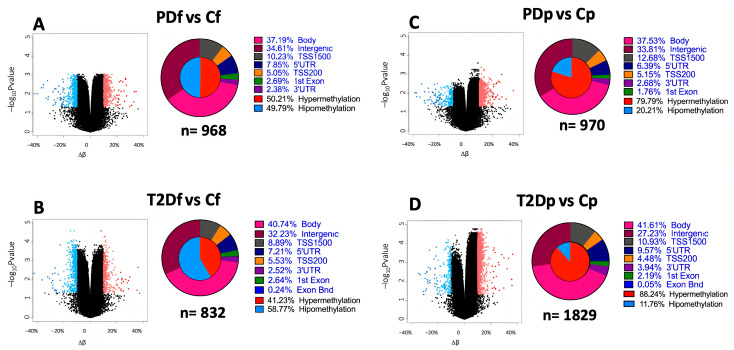
DNA methylation differences between groups by prandial state. (**A**,**B**) Fasting state. (**C**,**D**) Postprandial state. Differentially methylated CpG sites were selected based on adjusted *p*-value < 0.05 and |∆β|≥ 10%. Cf, control group-fasting; PDf, prediabetes group-fasting; T2Df, type 2 diabetes group-fasting; Cp, control group-postprandial; PDp, prediabetes group-postprandial; T2Dp, type 2 diabetes group-postprandial.

**Figure 3 ijms-26-11306-f003:**
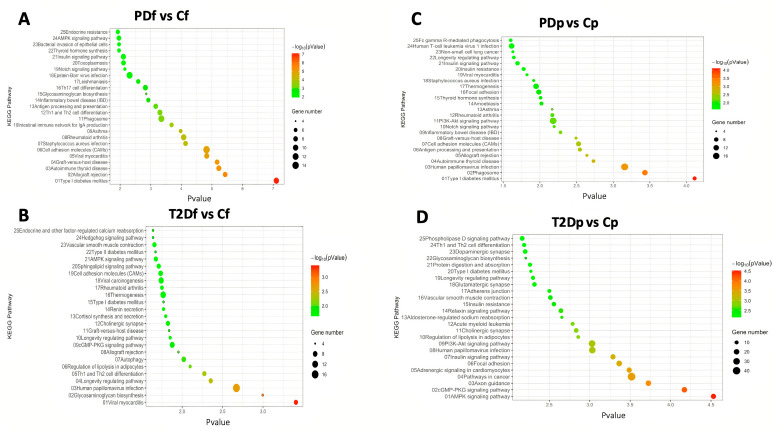
KEGG pathway enrichment analysis by prandial state. The top 25 pathways are shown. Dot size represents the number of genes in each KEGG pathway, and the color bar represents the *p*-value. (**A**,**B**) Fasting state. (**C**,**D**) Postprandial state. Cf, control group-fasting; PDf, prediabetes group-fasting; T2Df, type 2 diabetes group-fasting; Cp, control group-postprandial; PDp, prediabetes group-postprandial; T2Dp, type 2 diabetes group-postprandial.

**Figure 4 ijms-26-11306-f004:**
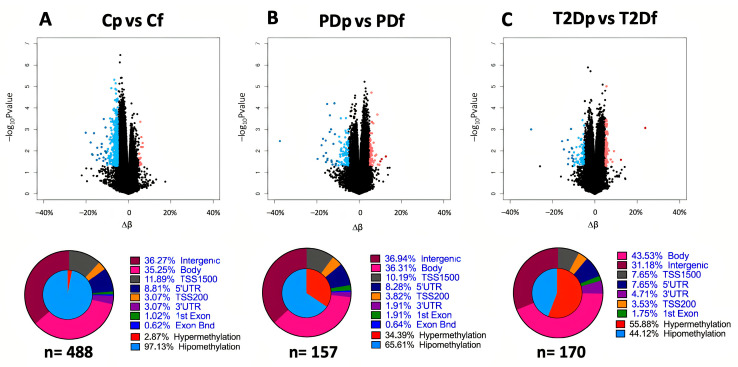
Volcano plots of the DNA methylation differences between prandial states in the control (**A**), prediabetes (**B**), and type 2 diabetes (**C**) groups. Select differentially methylated CpG sites with *p*-value < 0.05 and |∆β| ≥ 5% are presented in red (hypermethylation) and blue (hypomethylation). Cf, control group-fasting; PDf, prediabetes group-fasting; T2Df, type 2 diabetes group-fasting; Cp, control group-postprandial; PDp, prediabetes group-postprandial; T2Dp, type 2 diabetes group-postprandial.

**Table 1 ijms-26-11306-t001:** Clinical and cardiometabolic traits in the three study groups.

Clinical Traits (*n* = 29)	Control (*n* = 8)	Prediabetic (*n* = 9)	Type 2 Diabetes (*n* = 12)	*p*-Value ^†^
Woman/Man (*n*)	5/3	5/4	8/4	
Obesity (*n*)	0	3	9	
Hypertension (*n*)	0	4	6	
Age (years)	23 (21.2–24.7)	47 (26.7–56) *	49 (35.2–56.7) **	0.0063
Weight (kg)	60.8 (55.7–62.6)	75.7 (64–86)	83.5 (75–102.6) ***	0.0026
Waist circumference (cm)	71.3 (70.5–78)	95.6 (89.5–100.7) *	106 (87.7–117.7) ***	0.0005
Body mass index; BMI (kg/m^2^)	22.7(20.4–23.9)	28.9(27.6–31.3) **	34.6 (29.8–39.4) ***	<0.0001
Total fat (%)	23.8 (15.8–28.1)	37 (28.5–40.6) *	39.8 (35.3–45) ***	0.0008
Systolic pressure (mmHg)	103 (97.5–111)	119 (99.6–133.5)	116.3 (111–128.7) *	0.0493
Diastolic pressure (mmHg)	66 (62.5–69.7)	77 (64–80.5)	78.5 (75.2–86.2) **	0.0052
Glycated Hemoglobin; HbA1c (%)	4.8 (4.5–5)	5.2 (5.05–5.8)	6.1 (5.5–6.7) ***	0.0004
Fasting glucose (mg/dL)	82.5 (79–94)	110 (98–114.5)	153 (127.2–176) ****	<0.0001
2 h Glucose (mg/dL)	89 (81.5–104.2)	160 (146–178) *	225 (198.2–274.5) ***	<0.0001
Triglycerides (mg/dL)	83 (45–115.7)	116 (112–171.5)	172.5 (116.2–243.7) **	0.0048
Total cholesterol (mg/dL)	138 (111–151.8)	161 (150–209) *	185 (154.7–194) **	0.0086
HDL-cholesterol (mg/dL)	51 (37.2–65.5)	36 (34.5–46.5)	31.5 (26–40.7) **	0.0244
LDL-Cholesterol (mg/dL)	58 (54–88.5)	102 (84–142) *	109.5 (93–131.2) **	0.0110
VLDL-cholesterol (mg/dL)	16.5 (8.7–23.5)	23 (22.5–34)	34.5 (23.2–48.5) **	0.0054

All continuous values are shown as a median and interquartile range (IQR25 and IQR75). ^†^ Differences among groups were analyzed by Kruskall-Wallis; and differences between cases vs. control were analyzed using Dunn’s Post Hoc Test for joint ranks with the control group as control. *p*-values are represented using the following criteria: * for *p*-value < 0.05, ** for *p*-value < 0.01, *** for *p*-value < 0.001, and **** for *p*-value < 0.0001.

## Data Availability

The data set used and analyzed during the current study is available from the corresponding author on reasonable request.

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
