# Peer review of "Int. J. Mol. Sci.2025, 26(23), 11306;https://doi.org/10.3390/ijms262311306"

_ijms, 2025, doi:10.3390/ijms262311306_

Round 1

Reviewer 1 Report

Comments and Suggestions for Authors

The topic of this manuscript is novel, clinically relevant, and fits within the scope of the Journal. The use of paired adipose tissue biopsies is a significant strength.

However, several aspects should be addressed to ensure methodological rigor, interpretability, and reproducibility, particularly regarding:

Statistical thresholds and correction for multiple comparisons. The article uses two different criteria to declare CpGs as significant. In lines 366-367, a p-value <0.05 is reported, which appears to be unadjusted or corrected for multiple comparisons, potentially weakening the data.

Furthermore, the rationale for the postprandial test is unclear: why 3 hours? A justification citing relevant references is requested: could it be due to the insulin peak? Triglycerides? Are there previous publications using such a short interval for "fasting"?

Another important point is to delve deeper into the analysis of biological significance and translational relevance. It is considered necessary to strengthen the biological interpretation of the main highlighted genes, adding information about their role in metabolism, inflammation, insulin signaling, etc.

Finally, it is also suggested to review the consistency of the formatting. For example:

  • In some sentences, the % sign is used with a space between the number and the sign
  • One or two decimal places are used; this should be standardized.
  • The use of "p" in significances
  • Prentheses () or [] in quotation should also be consistent.

Furthermore, a general review of the language is suggested.

Comments on the Quality of English Language

The manuscript is written clearly in academic English. Minor grammatical, fluency, and stylistic corrections are recommended to improve readability and ensure consistency with scientific standards.

Author Response

Reviewer 1

We thank the reviewer for their comments; we believe they have significantly enhanced the quality of the article, and we hope it meets your expectations.

R1.C1.- Statistical thresholds and correction for multiple comparisons. The article uses two different criteria to declare CpGs as significant. In lines 366-367, a p-value <0.05 is reported, which appears to be unadjusted or corrected for multiple comparisons, potentially weakening the data.

Response: We appreciate the reviewer’s thoughtful comment regarding the statistical thresholds used to identify significant CpGs.  In this part of our study (section 2.3), we reported CpGs with a p-value <0.05, which corresponds to unadjusted values. We acknowledge that this approach may be considered less stringent; however, the use of unadjusted p-values is common in DNA methylation studies, as it allows the identification of potentially relevant CpGs that may otherwise be overlooked. We have now clarified in the manuscript that these p-values are unadjusted and made the following changes

  • Line 256, section 2.3, we added the word “unadjusted”, as follows: “We compared average β-values between prandial states using a less stringent cut-off than in previous comparisons (unadjusted p-value < 05 and |∆β| > 5%)”.
  • Discussion section line 523, we edited: “The sample size, even if adequate for an exploratory study, is relatively small, and the results may benefit from validation in larger cohorts.”

As follows: “The sample size, even if adequate for an exploratory study, is relatively small, and the results may benefit from validation in larger cohorts principally to identify alterations in DNA methylation after mixed-meal intake.”

R1.C2. Furthermore, the rationale for the postprandial test is unclear: why 3 hours? A justification citing relevant references is requested: could it be due to the insulin peak? Triglycerides? Are there previous publications using such a short interval for "fasting"?

Response: We thank the reviewer for this pertinent question regarding the methodological rationale for selecting the 180-minute postprandial time point for adipose tissue biopsies. Our justification rests on the following physiological considerations:

Following the consumption of a mixed meal, particularly one containing fat, plasma triglyceride concentrations commonly rise, reaching a peak between 3- and 5-hours post-ingestion [a, b]. This period reflects the peak circulation of chylomicrons, which are the primary vehicles for dietary fat transport. The 3-hour time point is strategically positioned within the early phase of this peak, representing a moment of high substrate availability for adipose tissue. At this time, insulin-stimulated lipoprotein lipase activity on the capillary endothelium of adipose tissue is maximal, promoting extensive hydrolysis of chylomicron-TGs and subsequent uptake of fatty acids into the adipocytes [c,d]. Therefore, a biopsy at 3-h allows for the direct assessment of adipose tissue responses—at both metabolic and transcriptional levels—to a maximal influx of diet-derived lipids [e]. This time point is therefore ideal for studying the integrated outcome of insulin signaling on gene expression and metabolic flux related to lipid storage and synthesis, which are central to our research question.

Finally, key studies have employed a similar time frame to investigate postprandial responses both in adipose and blood tissues. Eriksson et. al [c] utilized a 3.5-hour postprandial biopsy to study the regulation of adipose tissue lipoprotein lipase in subjects with type 2 diabetes and control subjects. Rask-Andersen et. al [f] studied DNA methylation and gene expression in whole blood before and 160 min after the ingestion of a standardized meal. Therefore, 3 hours represented a feasible time to obtain a biopsy within a relevant metabolic timeframe.

We have edited the main text as follows: 

line 534: “The GEMM family study [31] is a well-characterized multi-center cohort of symptom-free adults. Postprandial samples were obtained at 3-h after a liquid mixed-meal challenge (macronutrient composition: 65% carbohydrate, 15% protein, and 20% fat).”

By

line 534: “The GEMM family study [43] is a well-characterized multi-center cohort of symptom-free adults. Postprandial samples were obtained after a liquid mixed-meal challenge (macronutrient composition: 65% carbohydrate, 15% protein, and 20% fat). The biopsies were obtained after 3 hours, a time point selected based on previous findings suggesting that adipose tissue is metabolically most active approximately 3 hours post-caloric intake. [44,45]” 

References:

  1. Frayn, K. N. (2002). Adipose tissue as a buffer for daily lipid flux. Diabetologia, 45(9), 1201–1210.
  2. Karpe, F., Dickmann, J. R., & Frayn, K. N. (2011). Fatty acids, obesity, and insulin resistance: time for a reevaluation. Diabetes, 60(10), 2441–2449.
  3. Annuzzi G, Giacco R, Patti L, et al. Postprandial chylomicrons and adipose tissue lipoprotein lipase are altered in type 2 diabetes independently of obesity and whole-body insulin resistance. Nutr Metab Cardiovasc Dis. 2008;18(7):531–538.
  4. Eriksson JW, Buren J, Svensson M, Olivecrona T, Olivecrona G. Postprandial regulation of blood lipids and adipose tissue lipoprotein lipase in type 2 diabetes patients and healthy control subjects. Atherosclerosis. 2003;166(2):359–367.
  5. Morris C., O’Grada C.M., Ryan M.F., Gibney M.J., Roche H.M., Gibney E.R., Brennan L. Modulation of the Lipidomic Profile Due to a Lipid Challenge and Fitness Level: A Postprandial Study. Lipids Health Dis. 2015;14:65. doi: 10.1186/s12944-015-0062-x.
  6. Rask-Andersen, M., Bringeland, N., Nilsson, E. K., Bandstein, M., Olaya Búcaro, M., Vogel, H., Schürmann, A., Hogenkamp, P. S., Benedict, C., & Schiöth, H. B. (2016). Postprandial alterations in whole-blood DNA methylation are mediated by changes in white blood cell composition. The American journal of clinical nutrition, 104(2), 518–525. https://doi.org/10.3945/ajcn.115.122366

R1.C3.-Another important point is to delve deeper into the analysis of biological significance and translational relevance. It is considered necessary to strengthen the biological interpretation of the main highlighted genes, adding information about their role in metabolism, inflammation, insulin signaling, etc.

Response:

We thank the reviewer for this valuable suggestion to strengthen the biological interpretation and translational relevance of our findings. In the revised manuscript, we have expanded the discussion of the main highlighted genes by incorporating detailed information about their known functions in metabolism, inflammation, or their role in insulin signaling.

Line 335: ZNF714, which has emerged as a potential factor in metabolic diseases, as Crujeiras et al. reported differential DNA methylation in adipose tissue of individuals with insulin resistance, suggesting a potential role in the epigenetic regulation of metabolic pathways[24]. LCLAT1 (Lysocardiolipin acyltransferase 1), also known as ALCAT1, is a key enzyme in phospholipid remodeling that directly impacts mitochondrial function and oxidative stress, processes central to metabolic disease, which has recently been suggested to play a role in diabetes and obesity through epigenetic mechanisms [25,26]. Carnosine dipeptidase II (CNDP2) has been described as catalyzing the condensation reaction of lactic acid and phenylalanine, a metabolite that has been associated with the amount of adipose tissue in humans [27]. Individuals with obesity and type 2 diabetes have reported abnormal intragenic DNA methylation of HOOK2 when compared to individuals with non-diabetic conditions. The HOOK family is a group of cytoplasmic linker proteins associated with microtubules, possibly participating in GLUT4 translocation and glucose uptake [28].

R1.C4.-Finally, it is also suggested to review the consistency of the formatting. For example, In some sentences, the % sign is used with a space between the number and the sign One or two decimal places are used; this should be standardized. The use of "p" in significance. Parentheses () or [] in quotation should also be consistent.

Response: Thank you for your suggestion. We have carefully reviewed the manuscript to ensure consistency throughout the text.

R1.C5.- Furthermore, a general review of the language is suggested.

Response: The manuscript, especially the minor changes that were made to the current version, was carefully reviewed. It is important to note that the original version was previously revised by the professional editing service of San Francisco Edit (https://www.sfedit.net/about-san-francisco-edit/).

Reviewer 2 Report

Comments and Suggestions for Authors

This manuscript presents a study investigating fasting and postprandial DNA methylation signatures in subcutaneous adipose tissue from asymptomatic individuals with metabolic alterations, focusing on their implications for type 2 diabetes. The manuscript's objective is well-defined. It aims to investigate DNA methylation signatures in subcutaneous adipose tissue from asymptomatic, drug-naïve individuals with impaired glucose metabolism (prediabetes and type 2 diabetes) compared to individuals with normal glucose levels, in both fasting and postprandial states. This study was done to understand the epigenetic changes that occur in insulin-targeted tissues during the transition from fasting to postprandial states, with the goal of identifying early biomarkers for disease progression and metabolic disturbances. ​ This objective is significant, as it addresses a critical gap in understanding the epigenetic mechanisms underlying metabolic diseases, which could lead to improved prevention and personalized treatment strategies for prediabetes and type 2 diabetes. ​

This manuscript is an important study that correlates type 2 diabetes with epigenetic mechanisms. The following concerns have been highlighted to improve the manuscript:

1) The authors did not explain in detail why they selected DNA methylation signatures.
2) Did the authors compare their study with other diseases and metabolic changes?
3) Did the writers investigate the DNA sequence in their study?
4) The number of participants or subjects is small.

Author Response

Reviewer 2

We thank you for your comments; we believe they are very relevant and accurate. Here you can find the answer to each of them, which has allowed us to improve the quality of the manuscript. The revised version of the manuscript is available for your review.

R2.C1. The authors did not explain in detail why they selected DNA methylation signatures.

Response: Thank you for your comment. We improved the introduction section by exchanging the paragraph:

“Among these, DNA methylation is the most extensively studied epigenetic marker [8–12]. Altered DNA methylation and the corresponding changes in gene expression have been related to obesity and glucose impairment in cross-sectional and longitudinal studies [13–18]. Recent evidence suggests that methylation profiles in blood can differentiate between the fasting and postprandial states [19]. However, tissues targeted by insulin should be studied in-depth by explaining the differences among endophenotypes of patients with diabetes.”

By the paragraph:

line 86: “Among these epigenetic mechanisms, DNA methylation is one of the most extensively studied epigenetic markers [8–12]. Altered DNA methylation and the corresponding changes in gene expression have been related to obesity and glucose impairment in cross-sectional and longitudinal studies [13–18]. Our group previously identified several DNA methylation alterations in both adipose tissue and its mesenchymal stem cells during T2D, and these changes reverted after a clinical intervention [13–15]. Recent evidence suggests that methylation profiles in blood can differentiate between the fasting and postprandial states [19,20]. However, tissues targeted by insulin should be studied in-depth by explaining the differences among endophenotypes of patients with diabetes.”

We also incorporated the paragraph into the discussion section:

line 291: “Epigenetic regulation plays a relevant role in this process and can take place through various mechanisms, including DNA methylation, histone modifications, chromatin remodeling, and noncoding RNA modulation. Emerging evidence indicates that epigenetic regulation largely contributes to the occurrence and progression of multiple metabolic diseases [8]. Among the epigenetic mechanisms, DNA methylation has been suggested as a major driver in the environmental and epigenome intersection[21]. Furthermore, DNA methylation analysis using microarray technology in large cohort studies has generated substantial evidence linking methylation patterns to a wide range of diseases and phenotypic traits [22]. Together, these findings underscore the central role of DNA methylation as a robust and informative epigenetic marker for investigating metabolic disease risk.”

R2.C2. Did the authors compare their study with other diseases and metabolic changes?

Response: Thank you for your comment; it improved our discussion. As you may have noticed, there is a gap in the knowledge of the postprandial state and the modification of epigenetic marks. As far as we know, only two studies have addressed similar research, both of which are cited in the manuscript. Both studies examine epigenetic changes in blood in the postprandial state following caloric intake in individuals without metabolic disorders. (Pescador-Tapia A, et. al . Front Genet. 2021 May 7; Rask-Andersen M, et. al. Am J Clin Nutr. 2016). Comparing their as they name “relevant results” with our results, we identify five common genes (ABLIM1, ZNF516, SLC43A2, RPTOR, and S100P). Now, we have incorporated new information as follows:

Line 372 discussion section “In this same group of 130 DMCs, we also report five genes (ABLIM1, S100P, ZNF516, RPTOR, and SLC43A2) with previous reports of altered DNA methylation following caloric intake that also correlated with changes in expression levels [29]. S100P serum levels have been proposed as an indicator of peripheral neuropathy in type 2 diabetes [30].  ZNF516 has been implicated in the browning of adipose tissue [31]. In addition, RPTOR (Regulatory Associated Protein Of mTOR Complex 1) has been reported to play a key role in nutrient and insulin-sensing pathways regulating cell growth in cancer, while SLC43A2 has also been identified as a regulator of the mTORC1 complex in cancer studies [32]. Taken together, these findings indicate the need to investigate this signaling axis in non-cancer metabolic contexts, particularly in adipose tissue and its relationship with metabolic diseases.”

R2.C3. Did the writers investigate the DNA sequence in their study?

Response: The analysis of genetic variation in the patients is being carried out by another student as part of her PhD thesis. In this analysis, we have identified that the study population carries the expected percentages of ancestry for the Mexican Mestizo population. The detailed genetic analysis of the patients will be published in a separate publication.

R2.C4. The number of participants or subjects is small.

Response: Yes, we acknowledge that the sample size of our study is a limitation as declared in the manuscript. However, it is important to note that, based on a thorough review of the literature, many studies with substantially larger cohorts correspond to systematic reviews or epigenomic-wide association studies on blood cells rather than on adipose-related samples. For example, Lai CQ, J Lipid Res. 2016 Dec;57(12):2200-2207. doi: 10.1194/jlr.M069948. Epub 2016 Oct 24. PMID: 27777315. Si J, Circ Res. 2024 Oct 11;135(9):954-966. doi: 10.1161/CIRCRESAHA.124.325066.

Moreover, this trend is consistent with studies evaluating pre-and postprandial DNA methylation, which have also relied on comparable sample sizes and research designs. (Pescador-Tapia A, et al, Front Genet.) 2021 May 7;12:665769. doi: 10.3389/fgene.2021.665769. PMID: 34025721. Rask-Andersen M, et. al. Am J Clin Nutr. 2016 Aug;104(2):518-25. doi: 10.3945/ajcn.115.122366. Epub 2016 Jul 6. PMID: 27385611.)

We edited the discussion section as follows:

Line 519, discussion section: “The sample size, even if adequate for an exploratory study, is relatively small, and the results may benefit from validation in larger cohorts, principally to identify alterations in DNA methylation after mixed-meal intake.”

Reviewer 3 Report

Comments and Suggestions for Authors

The manuscript of “Fasting and postprandial DNA methylation signatures in adipose tissue from asymptomatic individuals with metabolic alterations” by Fabiola Escalante-Araiza and co-authors aims to analyze DNA methylation changes as biomarkers for early detection of type 2 diabetes progression and metabolic disturbances. The authors examined DNA methylation in adipose tissue samples from asymptomatic, drug-naive individuals with impaired glucose metabolism compared with individuals with normal blood glucose levels, both fasting and postprandial. They revealed that the control group formed a distinct cluster exclusively in the postprandial state, whereas the prediabetes and type 2 diabetes groups demonstrated overlapping patterns in both postprandial states, suggesting that individuals with impaired glucose metabolism may exhibit greater variability in DNA methylation patterns regardless of whether they are fasting or in the postprandial state. In general, DNA methylation increased in glucose-impaired groups, particularly in the postprandial state and in specific CpGs.

The topic of the study is highly relevant due to the sharp increase in the number of metabolic disorders and their complications worldwide and the urgent need to find new strategies for effective treatment. The author focused on a little-studied area of DNA methylation changes in SAT, highlighting the significance of the fasting and postprandial states in shaping the epigenetic landscape in individuals with metabolic alterations.

The study was conducted at a high experimental level. The manuscript as a whole and all its sections are well written and structured. All figures in the text properly show the data and are easy to interpret. The conclusions are consistent with the results obtained. The manuscript covers a large amount of literature data; the authors cited 35 articles, a significant portion of which were published in the last five years. All the references are appropriate.

The manuscript is well written and contributes to the systematization of new approaches to type 2 diabetes management. The study is very important for understanding the potential of epigenetic modifications in the form of DNA methylation temporal dynamics as therapeutic targets.

Minor

  1. The first sentence in the Introduction is overly complex and unclear. Please, rephrase.
  2. Epigenetic regulation is mediated by various mechanisms, including histone modification, chromatin remodeling, and modulation of non-coding RNA. Please, discuss the role of DNA methylation among these mechanisms in the Discussion section.

Author Response

Reviewer 3

We thank the reviewer for their time and effort in reviewing the article. Here you can see the responses to their two comments.

R3.C1. The first sentence in the Introduction is overly complex and unclear. Please, rephrase.

Response: Thank you for your comment. We changed the sentence as follows:

“Cardiovascular risk phenotypes of immunometabolic origin, such as insulin resistance, hyperinsulinemia, dysglycemia, dyslipidemia, and obesity, are major risk factors for type 2 diabetes (T2D).”

by

line 70: “Immunometabolic impairments such as insulin resistance, compensatory hyperinsulinemia, dysglycemia, dyslipidemia, and excess adiposity are well-established predictors of type 2 diabetes.”

R3.C2. Epigenetic regulation is mediated by various mechanisms, including histone modification, chromatin remodeling, and modulation of non-coding RNA. Please, discuss the role of DNA methylation among these mechanisms in the Discussion section.

We agree on this insightful comment. We have edited this in the discussion to a clearer contextualize DNA methylation within the broader landscape of epigenetic regulation:

Original sentence, line 291: “DNA methylation, an important epigenetic marker, plays a vital role in this process”

line 291: “Epigenetic regulation plays a relevant role in this process and can take place through various mechanisms, including DNA methylation, histone modifications, chromatin remodeling, and noncoding RNA modulation. Emerging evidence indicates that epigenetic regulation largely contributes to the occurrence and progression of multiple metabolic diseases [8]. Among the epigenetic mechanisms, DNA methylation has been suggested as a major driver in the environmental and epigenome intersection [21]. Furthermore, DNA methylation analysis using microarray technology in large cohort studies has generated substantial evidence linking methylation patterns to a wide range of diseases and phenotypic traits [22]. Together, these findings underscore the central role of DNA methylation as a robust and informative epigenetic marker for investigating metabolic disease risk.”